# Highly-Efficient Release of Ferulic Acid from Agro-Industrial By-Products via Enzymatic Hydrolysis with Cellulose-Degrading Enzymes: Part I–The Superiority of Hydrolytic Enzymes Versus Conventional Hydrolysis

**DOI:** 10.3390/foods10040782

**Published:** 2021-04-05

**Authors:** Karina Juhnevica-Radenkova, Jorens Kviesis, Diego A. Moreno, Dalija Seglina, Fernando Vallejo, Anda Valdovska, Vitalijs Radenkovs

**Affiliations:** 1Processing and Biochemistry Department, Institute of Horticulture, Graudu Str. 1, LV-3701 Dobele, Latvia; karina.juhnevica-radenkova@llu.lv (K.J.-R.); dalija.seglina@llu.lv (D.S.); 2Department of Environmental Science, University of Latvia, Jelgavas Str. 1, LV-1004 Riga, Latvia; jorens.kviesis@lu.lv; 3Phytochemistry and Healthy Foods Lab, Research Group on Quality, Safety and Bioactivity of Plant Foods, Department of Food Sciences and Technology, CEBAS-CSIC, University Campus of Espinardo, Edif. 25, 30100 Murcia, Spain; dmoreno@cebas.csic.es; 4Metabolomics Service, CEBAS-CSIC, University Campus of Espinardo, Edif. 25, 30100 Murcia, Spain; fvallejo@cebas.csic.es; 5Faculty of Veterinary Medicine, Latvia University of Life Sciences and Technologies, Kr. Helmana Str. 8, LV-3004 Jelgava, Latvia; Anda.Valdovska@llu.lv; 6Research Laboratory of Biotechnology, Latvia University of Life Sciences and Technologies, Strazdu Str. 1, LV-3004 Jelgava, Latvia

**Keywords:** recovery, biorefining, valorization, rye bran, ferulic acid, enzymatic hydrolysis

## Abstract

Historically *Triticum aestívum* L. and *Secale cereále* L. are widely used in the production of bakery products. From the total volume of grain cultivated, roughly 85% is used for the manufacturing of flour, while the remaining part is discarded or utilized rather inefficiently. The limited value attached to bran is associated with their structural complexity, i.e., the presence of cellulose, hemicellulose, and lignin, which makes this material suitable mostly as a feed supplement, while in food production its use presents a challenge. To valorize these materials to food and pharmaceutical applications, additional pre-treatment is required. In the present study, an effective, sustainable, and eco-friendly approach to ferulic acid (FA) production was demonstrated through the biorefining process accomplished by non-starch polysaccharides degrading enzymes. Up to 11.3 and 8.6 g kg^−1^ of FA was released from rye and wheat bran upon 24 h enzymatic hydrolysis with multi-enzyme complex Viscozyme^®^ L, respectively.

## 1. Introduction

One of the challenges facing food technology specialists at present is the utilization of agro-industrial by-products and food waste materials. Burgeoning demand for plant-made food products has resulted in a sharp increase in food processing waste, including cereal bran [1]. According to FAOSTAT (2020) [2], the world production of wheat (*Triticum aestívum* L.) and rye (*Secale cereále* L.) in the last five decades has increased by 190.5% from 257.4 million tons (Mt) in 1961 to 747.9 Mt in 2018. From the total volume of wheat and rye cultivated, roughly 85% or 635.7 Mt is utilized for flour production, while 10–15% or 74.8–112.1 Mt is converted into by-products, i.e., wheat and rye bran. These by-products are discarded or utilized rather inefficiently [3]. The limited value attached to bran as a food ingredient is associated with its structural complexity, i.e., the presence of cellulose, hemicellulose, and lignin which makes this material suitable mostly as a feed supplement, while in food production its use presents a challenge [1,4]. To overcome the adverse effect of by-products on the technological process and subsequently on quality of the final products, the use of biotechnological approaches through starch and non-starch polysaccharides (N-SPs) degrading enzymes has been proposed, which might be effective in partial hydrolysis of water-insoluble dietary fiber (N-SDF) of plant materials [5]. The modifications take place during enzymatic hydrolysis (EH), making these by-products act as selective growth-promoting substrates both for lactic acid bacteria and *Bifidobacteria* in the human intestinal tract [6,7,8] and, to a lesser extent, affect the sensory quality of the end-products [5]. Previous studies have shown that, along with the solubilization of N-SDF, phenolic acids are released. Ferulic acid (FA), in particular, is a compound that has received tremendous interest from researchers due to numerous benefits and feasible applications in many fields. The report of Zduńska et al. [9] reveals that FA has low toxicity and possesses such health benefits as anti-inflammatory, antimicrobial, anticancer (colon, skin, lung, and breast cancer cells), anti-arrhythmic, and antithrombotic activity. In an in vivo study, the beneficial effect of FA to alleviate lipid peroxidation in diabetic rats has been proven [10]. The favorable properties of natural FA have resulted in a sharp growth in the global market. The report of the Global Ferulic Acid Market, 2020 [11], suggests that the demand for FA is expected to increase at a compound annual growth rate CAGR of 6.0% and will reach ~90 M US$ globally by 2026 [12].

Ferulic acid (FA) 3-(4-hydroxy-3-methoxyphenyl)acrylic acid is hydroxycinnamic acid representative that is produced mostly by chemical synthesis at industrial-scale. The manufacturing of FA by this way is reported to be a laborious and time-consuming process (up to three weeks) that requires significant input of conditionally toxic reagents (malonic acid), solvents (piperidine), and catalysts (benzylamines) [13]. Moreover, this method allows for the production of FA as a mixture of *trans*- and *cis*-isomers, so additional purification process needs to be performed. Therefore, it may be beneficial to identify alternative sources and develop technologies for the release of FA that can be cost-effective and environmentally friendly. It has been proposed that under the proper conditions, hydrolysates which arise during EH of by-products might be purified and used for the production of FA that afterward could be utilized for the manufacturing of natural and inexpensive HPLC standards (a mixture of FA isomers or pure *trans*-FA), lipid oxidation inhibitors, agents with antimicrobial activities as medicines, and food additives [12,14,15].

In nature, FA appears as a component of plant cell-walls and forms a three-dimensional structure with cellulose, hemicellulose, and lignin [9]. The role of hydroxycinnamates lies in the regulation of cellular expansion and plant defense and reduction of the digestibility of cell walls by restricting access to carbohydrates [16]. Anson et al. [17] have found that FA in wheat bran appears both in free and bound form, where the free to bound ratio is 0.1:99.9 [18]. Conventional extraction using industrial extractants such as MeOH indicates strong binding affinity of FA to arabinoxylans, which limits the release of hydrocinnamate from the plant matrix [19,20]. The use of alkaline pretreatment was found to be effective for extraction of bound FA from various lignocellulosic biomass, including brewer’s spent grain [21]; however, concerning environmental pollution matters, production and utilization of strong bases or acids (for neutralization) would adversely affect human health, the environment, and industry [22]. In contrast to alkaline hydrolysis, the release of up to 95% of the total alkali-extractable FA could be achieved by means of EH with feruloyl esterase (FE) (EC 3.1.1.73) from *AspergiIlus niger* CBS 120.49 and the endo-1,4-β-xylanase (EC 3.2.1.8) from *Trichoderma viride* [23]. This assertion has been further reinforced by [24] who introduced a new biorefining process of corn residues and effective release of FA within 24 h. More recently, Wu et al. [8] observed a synergistic effect of xylanase and FE on the release of phenolic acids, in particular, FA during EH of wheat bran. The highest yield of FA that the authors recovered was 20 g per kg^−1^. Even though promising yields of FA were obtained, it should be mentioned that the production (specifically, the purification) of FE is a labor and material-costly process that makes this product applicable exclusively for research purposes and in vitro diagnostics.

The limited information on the recovery of FA from other cereal by-products, in particular from rye bran as well as for valorization purposes of this material, promoted the design of this study addressing the evaluation of the yield of FA from wheat and rye bran through biorefining process accomplished by C-DEs. The efficiency of FA release through hydrolytic enzymes will be compared with the conventional approach utilizing either alkali-assisted hydrolysis or micro-saponification. 

## 2. Materials and Methods

### 2.1. Plant Material

Two types of commercial food-grade bran samples harvested in 2019 were retrieved from the Latvian Mill Joint Stock Company (JSC) “Rigas dzirnavnieks” (Riga, Latvia), separated as summer wheat (*Triticum aestívum* L.) and rye (*Secale cereále* L.). Both brans consisted of outer pericarps, inner pericarps (cross cells, tube cells), seed coats (testa), hyaline, and aleurone layers with attached starch granules. The proximal analysis of the bran samples is shown in Table 1.

### 2.2. Plant Material Preparation for Alkaline and Enzymatic Hydrolysis and Analysis of Hydroxycinnamates

Each bran sample before EH was defatted following the methodology described by Radenkovs et al. [25] and ground to reach Ø 0.5 mm particle size using a variable speed rotor mill Pulverisette 14 (Fritsch, Idar-Oberstein, Germany). To exclude possible cell-wall disruption by native enzymes and microorganisms that can be localized on the surface of bran, the samples were mixed with double distilled water (DDW) at the ratio of 1:10 (*w*/*v*), loaded into 25 mL reagent bottles with screw caps (VWR™, International, GmbH, Darmstadt, Germany), and subjected to autoclaving using a digital autoclave with counter-pressure (Raypa, AES 110, Barcelona, Spain) for 10 min at 121 ± 1 °C temperature and counter pressure 2.0 Pa. Upon thermal processing, the liquid fraction was collected for further chromatographic analysis (the impact of thermal processing), while the solids were freeze-dried using a FreeZone freeze-dry system (Labconco, Kansas City, MO, USA) at −51 ± 1 °C under a vacuum of 0.065–0.070 mbar for 72 h. Dried samples were packed in polypropylene zip-lock silver bags (high-density polyethylene polymer, density 3 mm, Impak Co., Los Angeles, CA, USA) (200 g in each) and stored at a temperature of −18 ± 1 °C until further analysis and use, a maximum of 4 wk. Moisture content was analyzed gravimetrically as proposed by Ruiz [26].

### 2.3. Chemicals and Reagents

Phenolic compound commercial standards, i.e., caffeic acid (CA), a mixture of *trans*- and *cis*-isomers of ferulic acid (FA), vanillic acid (VA), and *p*-coumaric acid (*p*-CA) were purchased from Sigma-Aldrich Chemie Ltd., (Steinheim, Germany). Ethylenediaminetetraacetic acid disodium salt 2-hydrate (Na_2_EDTA) was acquired from Panreac Quimica S.L.U. (Barcelona, Spain). Hydrochloric acid (HCl), formic acid (CH_2_O_2_) (puriss r.a.), sodium hydroxide (NaOH), potassium hydroxide (KOH), citric acid (C_6_H_8_O_7_), sodium citrate dihydrate (C_6_H_5_Na_3_O_7_·2H_2_O) of reagent grade, methanol (MeOH), ethanol (EtOH), and acetonitrile (MeCN) of UHPLC grade were purchased from Merck KGaA (Darmstadt, Germany). The ultrapure water was produced using the reverse osmosis technique Mili-Q^®^ Direct water purification system (Millipore, Bedford, MA, USA). 

### 2.4. Enzymes

Industrial starch and N-SPs degrading enzymes preparations have been kindly provided by the company “Novozymes^®^” (Bagsvaerd, Denmark) for laboratory purposes. Three types of starch and N-SPs degrading enzymes, i.e., Amylase^®^ AG XXL (AMY), Celluclast^®^ 1.5 L (CELLU), Viscozyme^®^ L (VISCO) were used to hydrolyze carbohydrates. High purity recombinant FE from rumen microorganism was obtained from Megazyme™ (Bray, Ireland) to liberate esterified FA forms from plant material matrix during an additional hydrolysis process. In this work, enzymes were tested both independently and in combination, in order to establish the synergetic effect. The list of enzymes used is given in Table 2.

### 2.5. Hydrolysis of Wheat and Rye Bran

#### 2.5.1. Alkaline-Assisted Hydrolysis

To estimate the total amount of bound FA (ester-linked), the alkali-assisted hydrolysis of bran samples was done according to the method proposed by Zhang et al. [27] as a control. In brief, 1 g of bran was mixed with 10 mL 2 M NaOH in 100 mL reagent bottles with screw caps (VWR™), followed by incubation in a water bath at 60 ± 1 °C for 4 h at 85 rpm (Memmert^®^, Büchenbach, Germany). Afterward, the mixture was subjected to ultrasound treatment at 42 kHz with output wattage of 100 W for 10 min at 25 ± 1 °C using a “Branson 3510” ultrasonic bath (Branson Ultrasonics, Danbury, CT, USA), followed by pH adjustment with 4 M HCl to reach pH 4.0 ± 0.2 and centrifuged at 20,160× *g* at in an “EBA 21 Hettich” centrifuge (Hettich Zentrifugen, Tuttlingen, Germany) for 10 min at 25 ± 1 °C. Following centrifugation, the supernatants were decanted and 3 times extracted with 3 volumes of ethyl acetate:diethyl ether (1:1, *v*/*v*). The organic layer was collected in a pear-shaped flask and vacuum dried using a Laborota 4000 vacuum rotary evaporator (Heidolph, Schwabach, Germany). The yellowish powder was then re-dissolved in 10 mL of 70% acidified MeOH (methanol:DDW:formic acid ratio 70:24:1 *v*/*v*/*v*). The collected extracts were filtered through a polyvinylidene fluoride membrane filter (PVDF) with a pore size of 0.22 µm (Agela Technologies™, Torrance, CA, USA) before analysis. Filtered samples were diluted appropriately before HPLC-DAD-ESI/MS^n^ and UPLC-ESI-QTOF/MS analysis. 

In this experiment, a micro-saponification of wheat and rye bran by methanolic KOH was performed for de-esterification of bound FA. Briefly, 1 g of bran was mixed with 10 mL 1 M KOH in 80% MeOH (methanol:DDW ratio 80:20 *v*/*v*) in 25 mL reagent bottles with screw caps (VWR™) followed by 1 min intensive Vortexing using an “IKA TTS2” mixer (IKA^®^-Werke GmbH & Co. Kg, Staufen, Germany) and incubation in a water bath at 60 ± 1 °C for 1 h at 85 rpm. After incubation, the samples were subjected to ultrasonication for 10 min at 25 ± 1 °C, followed by pH adjustment with 4 M HCl to reach pH 4.0 ± 0.2 and centrifugation at 25 ± 1 °C for 10 min at 20,160× *g*. The obtained hydrolysates before chromatographic analysis were filtered using 0.22 µm PVDF.

#### 2.5.2. Enzyme-Assisted Hydrolysis

Step I. The EH of wheat and rye bran employing hydrolytic enzymes was done in a water bath with a capacity of 22 L and a horizontal shaking system and thermostatic and temporal control system. To estimate the influence of the treatment time on the yield of hydroxycinnamates (FA specifically), the process of EH was done within the range of 4–24 h depending on the enzyme used. The optimal conditions for each enzyme were selected individually based on the Novozymes^®^ and Megazyme™ recommendations and according to the protocols described by [28,29]. The EH of N-SPs, i.e., cellulose and hemicellulose was accomplished utilizing either multi-enzyme complex VISCO or CELLU.

For this purpose, 1 mL 0.5 M sodium citrate buffer containing either 6 FBG mL^−1^ of endo-1,4-β-xylanase (VISCO) or 10 EGU mL^−1^ of 1,4- β-d-endoglucanase (CELLU) was added to 100 µg of bran sample. The mixture was then Vortexed for 2 min and incubated in a water bath at 44 ± 0 °C for 24 h, at 85 rpm, respectively. The release of FA at different time points, i.e., 4, 8, 12, and 24 h of enzymatic hydrolysis, was ascertained chromatographically. 

Before chromatographic analysis to terminate the reaction, the obtained hydrolysates were subjected to ultrasonication at 42 kHz for 10 min at 25 ± 1 °C, followed by centrifugation of 20,160× *g* for 10 min at 25 ± 1 °C. Before analysis, the supernatants were filtrated through 0.22 µm PVDF membrane filters. The kinetic of hydrolytic enzymes was estimated by the selective measurement of glucose level in hydrolysates using a commercial blood glucose meter (Contour^®^Plus, Bayer Healthcare AG, Leverkusen, Germany) as proposed by Heinzerling et al. [30].

Step II. A set of experiments was done to estimate the synergistic effect of sequential hydrolysis of bran samples utilizing either amylolytic enzymes alone or in combination with N-SPs degrading enzymes.

For destarching purposes, AMY (highly stable across a wide range of pH values) isolated from *Aspergillus niger* was used. The selected enzyme does not require additional alkalization of the material, which makes it attenable to maintain constant pH of substrate for both cellulolytic and amylolytic enzymes. In brief, 100 µg of bran sample was mixed with 1.0 mL 0.1 M sodium citrate buffer (pH 4.6), containing 10 AGU mL^−1^ of glucan-1,4-*α*-glucosidase (AMY) and incubated either for 1 or 4 h in a water bath at 55 ± 1 °C under constant agitation of 85 rpm. To terminate the reaction, the slurry was then subjected to ultrasonication at 42 kHz for 10 min at 25 ± 1 °C using an ultrasonic bath. Thereafter, an aliquot of 300 µL was collected and centrifuged at 20,160× *g* for 10 min at 25 ± 1 °C. The supernatants collected were then filtrated through 0.22 µm PVDF membrane filter. The amount of FA and other hydroxycinnamates was quantitatively and qualitatively evaluated by using the above-mentioned systems. The following EH process for 4 and 24 h was done using destarched slurry as a substrate for endo-1,4-β-xylanase (VISCO) and 1,4- β-d-endoglucanase (CELLU). The amount of enzymes added corresponded to 6 FBG mL^−1^ and 10 EGU mL^−1^ in 0.1 M sodium citrate buffer.

Step III. To test the synergistic effect of endo-1,4-β-xylanase (VISCO) and FE (Megazyme™), the sequential EH of wheat and rye bran was accomplished with the use of defatted, destarched slurry with the following determination of FA in hydrolysates (Figure 1). The release of FA was done either with FE alone or after EH with VISCO. To initiate the catalysis reaction of 4-hydroxy-3-methoxycinnamoyl (feruloyl) groups between FA and arabinose, the pH of substrate was adjusted to 7.0 ± 0.2 by adding 1 mL of 0.2 M MOPS buffer solution (pH 7.0 ± 0.2), containing 0.04% (*w*/*v*) Na_2_EDTA, 0.02% (*w*/*v*) sodium azide, and 7.5 U mL^−1^ (20 µL) of FE [16]. The yield of FA upon 4 to 24 h was determined.

### 2.6. Generation of Standard Curves

Quantification of compounds was done using calibration curves of the following standards: CA, FA, VA, *p*-CA. All calibration curves had R^2^ > 0.98.

### 2.7. The HPLC-DAD-ESI/MS^n^ Conditions 

Quantitative analysis of FA was done using an Agilent HPLC 1200 series system (Agilent Technologies, Waldbronn, Germany) as previously described by Ferreres et al. [31]. The HPLC consisted of a binary pump (model G1376A), an autosampler (model G1377A) equipped with a thermostat module (G1330B), a micro vacuum degasser (model G1379B), and a diode array detector (model G1315D). The HPLC system was controlled by ChemStation software (Agilent, v. B.01.03- SR2). Chromatographic separation was done on a reversed-phase Fortis UniverSil HS C18 column (4.6 × 250 mm; 5 μm; Fortis™ Technologies Ltd. Cheshire, UK). The flow rate of the mobile phase was 0.8 mL min^−1^ at room temperature (25 °C), with the elution gradient being 1.0% formic acid (solvent A) and pure ACN (solvent B). A stepwise gradient elution program from 15% solvent B up to 20% in 25 min, from 25 to 55 min solvent B concentration was increased to 70% and held for 5 min. The injection volume was 5 μL. Spectral data from all peaks were accumulated in the range of 240–400 nm, and chromatograms were recorded at 320 and 280 nm. For the identification of hydroxycinnamates, a Bruker series LC/MSD HCT Ultra ion trap detector with electrospray ionization (ESI) interface (Agilent, v. 6.1) was used. The ionization conditions were adjusted at 350 °C and 4.0 kV for capillary temperature and voltage, respectively. The nebulizer pressure and flow rate of nitrogen (N_2_) (purity–99.0%) were 65.0 psi and 11 L min^−1^, respectively. The full scan mass covered the range from 100 up to 1200 atomic mass units (AMU), with target mass of 400 AMU. Collision-induced fragmentation was performed in the ion trap using helium (He, purity–99.0%) as the collision gas, with voltage ramping cycles from 0.3 up to 2 V. Mass spectrometry data were acquired in the negative ionization mode. MS^2^ was carried out in the automatic mode on the more abundant fragment ion in MS.

### 2.8. The UPLC-ESI-QTOF/MS Conditions

The analyses were carried out using an Agilent 1290 Infinity series LC system coupled to a 6550 I-Funnel Accurate-Mass QTOF (Agilent Technologies, Waldbronn, Germany) with a dual electrospray ionization interface (ESI-Jet Stream Technology) for simultaneous spraying of a mass reference solution that enabled continuous calibration of detected *m*/*z* ratios. Sample of 1 μL was injected onto a reversed-phase Luna Omega column (1.6 μm, PS C18, 100 Å, 50 × 2.1 mm; Phenomenex, Macclesfield, UK) with SecurityGuard ULTRA cartridges of the same material operating at 30 °C and a flow rate of 0.5 mL min^−1^. The mobile phases used were acidified DDW (0.1% formic acid) (A) and acidified ACN (0.1% formic acid) (B). Separation of compounds was done using the following gradient conditions: elution started with 1% B to obtain 10% B at 8 min, 30% B at 14 min, 50% B at 16 min, and 1% B at 20 min. Furthermore, MeOH injections were included every three samples as a blank run to avoid the carry-over effect.

Data were acquired using the Mass Hunter Workstation software (version B.08.00, Service Pack 1, Agilent Technologies, Santa Clara, CA, USA). The system was operated using both negative and positive ion polarity, and data were acquired in centroid and profile mode, with a data storage threshold of 5000 absorbance for MS. The operating conditions were as follows: gas temperature of 280 °C, drying nitrogen gas of 9 L min^−1^, nebulizer pressure of 45 psi, sheath gas temperature of 400 °C, sheath gas flow of 12 L min^−1^, a capillary voltage of 3500 V, nozzle voltage of 500 V, fragmentor voltage of 100 V, skimmer of 65 V and octopole radiofrequency voltage of 750 V. TOF spectra acquisition rate/time was 1.5 spectra s^−1^ and 666.7 ms/spectrum, respectively, and transients/spectrum were 5484. The mass range was between 50 and 1100 *m*/*z*. At the beginning of the batch, the instrument was calibrated to assure mass accuracy during the MS analysis using a mixture of reference compounds (Tuning Mix). Furthermore, continuous internal calibration was performed during analyses using signals *m*/*z* 112.9855 and *m*/*z* 1033.9881 in negative polarity and *m*/*z* 121.0509 and *m*/*z* 922.0098 in positive polarity. Auto recalibration reference mass parameters were a detection window of 100 ppm and a minimum height of 1000 counts. Data were processed using the Mass Hunter Qualitative Analysis software (version B.08.00, Service Pack 1, Agilent Technologies, Santa Clara, CA, USA).

### 2.9. Scanning Electron Microscopy (SEM)

The morphology of untreated control and EH bran was analyzed by SEM using a Tescan Mira/LMU scanning electron microscope (Brno-Kohoutovice, Czech Republic) according to the method proposed by Rahman et al. [3]. The conditions were set up to operate under low vacuum mode (ESEM, PH_2_O = 1.0–4.8 torr) (PH_2_O = 0.1–1.0 torr) using a large field detector (LFD). For ESEM analysis, samples were mounted onto SEM stubs using double-sided adhesive carbon discs and structural changes were observed at 10–20 kV acceleration voltage. To improve sample electrical conductivity and ability to reflect electrons and hence ensure higher resolution of images, all samples were covered with a 15 nm thick gold layer before analysis.

### 2.10. Statistical Analysis

The results obtained are shown as means ± standard error of the mean from three replicates (*n* = 3). The *p*-value < 0.05 was used to denote significant differences between mean values determined using one-way analysis of variance (ANOVA) and the Duncan’s multiple range test done using the assistance of IBM^®^ SPSS^®^ Statistics program 20.0 (SPSS Inc., Chicago, IL, USA).

## 3. Results and Discussion

### 3.1. Release of FA Using Conventional Alkaline-Assisted Hydrolysis

Wheat and rye bran as a by-product of grain processing have complex structures, where starch and non-starch polysaccharides, proteins, lipids, and other high-molecular compounds are present as highly branched and cross-linked polymers. Complexity in structure makes the recovery of FA through conventional extraction challenging. Additional pre-treatment aimed at gentle destruction of cell-walls is of great interest. Among the organic solvents and pre-treatment types used, alkaline hydrolysis was shown to be an effective tool for the extraction of hydroxycinnamates (FA specifically) from agro-industrial by-products, including wheat bran [19,20,23,32]. This finding has been reinforced by the current study. Pre-treatment of wheat and rye bran either with 2 M NaOH or 1 M KOH in 80% MeOH delivered 2.13 to 11.4 g kg^−1^ DW of FA, respectively (Figure 2A). The yield of FA obtained, however, is significantly (*p* < 0.05) higher than that reported by Ideia et al. [21]. It is worth noting, though, that at the same time as FA release, the formation of 4-vinylguaiacol (4VG) (Figure 2B) and *p*-vinylphenol (*p*-VP) (Figure 2C) as the first degradation products of FA and *p*-CA was observed. The presence of these compounds was confirmed by the UPLC-ESI-QTOF/MS system.

The intensities of peaks indicate the relative abundance of these two compounds in alkali-hydrolyzed samples, the percentage ratio of FA to 4-VG and *p*-CA to *p*-VP corresponds to 81.0:19.0 and to 45.2:54.8, respectively. This finding supports the conclusion that the formation of 4-VG (experimental *m*/*z* 196.0611) and *p*-VP (experimental *m*/*z* 119.0502) might take place not only during the pyrolysis and microbiological conversion via decarboxylation pathway as reported by Ohra-Aho et al. [33] and Salgado et al. [32], but also during mild alkaline hydrolysis, leading to degradation compounds of interest. Moreover, considering the environmental pollution matters, the utilization of base as a pre-treatment type is not environmentally feasible since this process requires the use of strong bases and acids which significantly endanger both the health of personnel involved in the production of FA and the environment, hence alternative and gentle biorefining processes of the material need to be identified.

### 3.2. Release of FA Using Enzyme-Assisted Hydrolysis with Commercial Cellulolytic Enzymes 

Very recently, in an attempt to effectively recover FA, the sequential hydrolysis of wheat bran biorefining has been developed by an Italian research group [34]. An observation made by the authors revealed the increase of FA followed EH with commercially available hydrolytic enzymes, i.e., Alcalase^®^ (protease) Termamyl^®^ (α-amylase), and Peptopan^®^ (fungal xylanase). The yield of FA using this approach was somehow similar to that found in this study applying alkaline-hydrolysis with 2 M NaOH for 1 h. The amount of FA recovered was 0.82–1.05 g kg^−1^ of bran. 

To validate the ability of N-SPs degrading enzymes alone or in combination with glycoside hydrolase (AMY) and FE (Megazyme™) to release the bound FA forms, a serial EH of wheat and rye bran was applied. In this set of experiments, the wheat and rye bran samples underwent EH for 4 to 24 h though N-PSs, i.e., CELLU and VISCO to solubilize cellulose and hemicellulose. As is seen, the highest concentration of FA in both cases using either CELLU or VISCO yielded after 24 h with enzyme doses of 10 EGU and 6 FBG mL^−1^, respectively (Figure 3). Considering the yield of FA using alkali-assisted hydrolysis with methanolic 1 M KOH in 80% MeOH for 1 h, the recovery of bound FA from wheat and rye bran using CELLU as a sole enzyme was 58.8 and 51.7%, respectively. In turn, 75.4 and 135.6% of the total alkali-extracted FA was released from wheat and rye bran by the action of endo-1,4-β-xylanase and α-L-arabinofuranosidase present in multi-enzyme complex VISCO, respectively. Notably lower release of FA is due to the presence of higher Klason lignin content in wheat bran than that of rye, which is reported to act as a physical barrier obstructing the access of hydrolytic enzymes to hemicellulose and cellulose. It is worth noting that the degradation of FA to a lesser extent was observed using EH, where the percentage ratio of FA to 4-VG corresponded to 91.0:9.0. In turn, no presence of *p*-CA degradation products was found in the hydrolysates obtained upon EH. The superiority of VISCO over the other cellulolytic enzymes was highlighted by Mahmoudi et al. [35], indicating the better recovery of total phenols and flavonoids from enzymatically pre-treated sweet basil (*Ocimum basilicum* L.) leaves and their hydrodistilled residue by-products. 

### 3.3. Release of Glucose Using Enzyme-Assisted Hydrolysis with Commercial Cellulolytic Enzymes 

The main substrates for the cellulolytic enzymes used in this study were those non-starch polysaccharides present in wheat and rye bran. Since the main activity of hydrolytic enzymes used is cellulolytic, in addition to the FA content, the efficiency of hydrolysis was estimated by measuring the amount of glucose in hydrolysates. The results of glucose content show values ranging between 11.23 and 31.20 mmol L^−1^ for wheat and rye bran hydrolysates after EH with CELLU, with rye bran having the highest glucose level, with wheat bran the lowest (Figure 3). The maximum yield of glucose was released after 24 h of EH. The same tendency of glucose increase has been observed for wheat and rye bran samples upon 24 h EH with multi-enzyme complex VISCO, where the highest values corresponded to 44.14 and 49.50 mmol L^−1^ for wheat and rye bran hydrolysates, respectively. The use of EH with CELLU led to up 2.4- and 2.6-fold increase in glucose content as compared to initial glucose level after EH for 4 h, while the amount of glucose was up to 1.8- and 1.9-fold higher after VISCO 24 h treatment, indicating superior ability of CELLU to catalyze glycosidic bonds within glucose chains in cellulose within the first hours of EH. The results of glucose content in wheat bran hydrolysates were shown to have a strong and moderate positive correlation with FA in the following order: CELLU 10 EGU (*r* = 0.9887) > VISCO 3 FBG (*r* = 0.9117) > CELLU 5 EGU (*r* = 0.8357) > VISCO 6 FBG (*r* = 0.7642). Similar results were obtained carrying out correlation analyses of glucose level with FA in rye bran hydrolysates, although with a different order of correlations: CELLU 10 EGU (*r* = 0.9804) > VISCO 6 FBG (*r* = 0.9744) > VISCO 3 FBG (*r* = 0.9524) > CELLU 5 EGU (*r* = 0.8017). The results show that since VISCO is a multi-enzyme complex containing a wide range of carbohydrases, i.e., cellulases, arabanases, β-glucanase, hemicellulase, and xylanases where the end products of synergistic action could be glucose, arabinose, mannose, galactose, and xylose, the release of FA is to a lesser extent associated with the accumulation of glucose. This finding is reinforced by report of Shin et al. [24], pointing to the strongest correlation of the release of FA with the liberation of reducing sugars, in particular glucose, xylose, and arabinose from corn bran upon EH with a crude hemicellulase. The amount of above-mentioned saccharides after EH with VISCO needs to be considered. This observation makes it possible to also confirm that FA in both wheat and rye bran is linked to arabinofuranosidase residues in arabinoxylans, while continuous EH with L-arabinofuranosides and endo- 1,4-β-xylanase presented in VISCO results in the solubilization of W-I arabinoxylans and the depolymerization of W-S arabinoxylans with the simultaneous release of FA [1]. The amount of FA was found to be significantly (*p* < 0.05) lower in hydrolysates after EH with CELLU, while the strongest correlation between FA and glucose indicates that partial solubilization of arabinoxylans could be achieved even through the application of cellulose-degrading enzyme CELLU as a sole enzyme [36].

### 3.4. Structural Changes in Wheat and Rye Bran Morphology Induced by Cellulolytic Enzymes 

The decomposition of bran N-SPs caused by EH was confirmed by SEM (Figure 4). After 24 h of EH with multi-enzyme complex VISCO obvious signs of epidermal cracking (holes size of 20–30 μm) were seen on the surface of the wheat bran sample (Figure 4D1). The EH for 4 h led to partial degradation of the epidermal layer of wheat bran and opening cellulose microfibers as depicted in Figure 4E1, however, to a lesser extent than after 24 h of EH (Figure 4D1). As seen, more tangible degradation of the epidermal layer (Figure 4F1) with clearly observed degraded aleurone layer and empty spaces the places where cellulose microfibrils shall be located. Degradation of cellulose and hemicellulose of the wheat and rye bran (data not shown) is due to the ability of cellulolytic enzymes act specifically on 1,4-β-d-glycosidic linkages in cellulose and on 1,2-α-, 1,3-α-, and 1,5-α-L-arabinofuranosidic and 1,4-β-d-xylosidic linkages in arabinoxylan, arabinan, and other arabinose-containing hemicelluloses, releasing relatively shorter polysaccharides or oligosaccharides composed either of glucose or xylose [1]. 

In general, EH has led to a decrease of total dietary fiber combined with a shift from water-insoluble fiber (W-I) to water-soluble (W-S). The results are consistent with those observed by Arrigoni et al. [37], noting the change in the ratio of W-I to W-S upon EH with VISCO, the percentage distribution of two fractions being 65:35 for apple pomace, while 73:27 for sugar beet pulp, causing decreases of 20% and 9% of total fiber, respectively.

### 3.5. Release of FA Using Enzyme-Assisted Hydrolysis Accomplished by Glycolytic and Cellulolytic Enzymes 

For the specific release of FA from wheat and rye bran cell-wall matrix, commercial FE and AMY were tested alone or in combination with CELLU and VISCO enzymes. Before EH with cellulolytic enzymes, wheat and rye bran samples were destarched for 1 or 4 h with 10 AGU mL^−1^ of AMY at 55 °C. At the end of the process, destrached slurry was subjected to ultrasonication for 30 min to terminate the catalytic reaction. Afterward, the CELLU or VISCO in the dose of 10 EGU and 6 FBG mL^−1^, respectively, was separately added followed by EH for 4 or 24 h. Both combinations were assayed for FA release.

An earlier study of Yu et al. [38] showed that a combination of acidic pre-treatment, amylolytic and cellulolytic enzymes under sequential hydrolysis might be applicable for the release of more phenolic acids, including FA from barley matrix, pointing to the ability of α-amylase acts on ester linkages between sugars and FA. This statement has not been confirmed in the current research, as the presence of free FA was detected as traces in hydrolysates either after acid hydrolysis alone or acid pre-treatment in combination with enzymatically-assisted hydrolysis with AMY (Figure 5 and Figure 6).

This was not surprising, since α-amylase acts specifically on the 1,4-α-d-glycosidic linkages between glucose monomers in amylose and amylopectin, converting these polymers into simple sugars such as glucose, maltose, and maltotriose [29]. Since both wheat and rye bran represent a significant amount of starch (Table 1), this step was done only to make the substrate area more accessible for cellulolytic enzymes rather than to cleave ester linkages between FA and arabinose as speculated by Yu et al. [38]. A combination of AMY with VISCO (Figure 6D) delivered a considerably lower amount of FA than when VISCO was used as a sole enzyme. However, a synergistic effect was observed following EH by AMY with CELLU for 4 and 24 h, respectively (Figure 6D). Up to 45.3 and 51.6% of the total alkali-extractable FA was released from wheat and rye bran using this combination, respectively. The notably lower yield of FA after EH with AMY + VISCO is presumably due to the reduction of catalytic activity of endo-1,4-β-xylanase in VISCO within further EH caused by AMY α-amylase inhibitory activity [39].

### 3.6. Release of FA Using Enzyme-Assisted Hydrolysis with Cellulolytic Enzymes in Combination with Feruloyl Esterase

As can be seen, the highest concentration of FA by FE as a sole enzyme was reached after 24 of hydrolysis when the dose of the enzyme was 7.5 U mL^−1^ (Figure 6A,B). The release of up to 33.7% and 47.2% of the total alkali-extractable FA was achieved by EH of wheat and rye bran with FE, respectively. The results obtained are consistent with those reported by Benoit et al. [16], while considerably higher than that of Ferri et al. [34]. Differences in the amounts of FA release could be associated with the types of FE used.

The synergistic effect of FE coupled with cellulolytic enzyme VISCO was not observed as compared to the yield of FA when the VISCO was used as a sole enzyme. The highest yield of FA in this experiment was reached after 24 h of EH by FE applying pre-treatment with VISCO for 24 h (Figure 6C). The 72.4 and 92.4% of the total alkali-extractable FA was released from wheat and rye bran, respectively. The lower yields of FA are presumably due to partial hydrolysis of free FA caused by extended temperature and light exposure for 4 and 24 h during EH with FE [40]. The percentage ratio of FA to 4-VG acquired by UPLC-ESI-QTOF/MS system corresponded to 73.0:27.0 and to 77.4:22.6 for wheat and rye bran, respectively. Besides, additional energy and resource input make this process not economically feasible.

## 4. Conclusions

The present study was undertaken to establish the ability of hydrolytic enzymes to act on 1,4-β-d-glycosidic linkages in cellulose and on 1,2-α-, 1,3-α-, and 1,5-α-L-arabinofuranosidic and 1,4-β-d-xylosidic linkages in arabinoxylans of wheat and rye bran and release bound ferulic acid (FA) from lignocellulosic biomass. For these purposes, two commercial food-grade types of grain by-products, i.e., wheat and rye bran, were used as substrates for hydrolytic enzymes. The process of enzymatic hydrolysis (EH) was done by sequential biorefining, applying four hydrolytic enzymes. The EH was performed under optimal conditions for each enzyme and the release of FA was monitored over 24 h. Among the enzymes tested, the superiority of multi-enzyme complex Viscozyme^®^ L has been highlighted, as the maximal yield in the amount of 11.3 and 8.6 g kg^−1^ FA from rye and wheat bran was obtained upon 24 h of EH, respectively. Up to 75.4 and 135.6% release of the total alkali-extractable FA was achieved from wheat and rye bran, respectively. No synergistic effect was observed combining Viscozyme^®^ L either with feruloyl esterase or Amylase^®^ AG XXL. When combining Celluclast^®^ with Amylase^®^ AG XXL up to the 1.1-fold and 0.9-fold increase in the yield of FA was found as compared to Celluclast^®^ when used alone.

The UPLC-ESI-QTOF/MS analysis of enzymatically obtained hydrolysates with Viscozyme^®^ L confirmed the presence of degradation products of FA and *p*-coumaric acid to a lesser extent than that of alkaline-hydrolysis, indicating gentler decomposition of bran matrix and fewer negative impact on compounds of interest. In the process of enzymatic hydrolysis, the formation of glucose as the final product of cellulose depolymerization was observed in this study. In the context of the zero-waste biotechnological process, the glucose arisen could be further exploited for bioethanol production, while solid residues may find applicability within the food and agricultural sector as a material suitable for synbiotic food products and compost production.

To summarize, the introduced approach is an environmentally friendly alternative with a safer profile than a conventional one and could represent the future for sustainable industrial-scale ferulic acid production. However, the issue of efficient and environmentally friendly purification of ferulic acid remains relevant and needs to be addressed. Part II of this series of studies will discuss the purification of ferulic acid taking advantage of solid-phase extraction techniques by commercially available polymers such as AmberLite™, Sephadex™, and polyvinylpyrrolidone (PVP).

## Figures and Tables

**Figure 1 foods-10-00782-f001:**
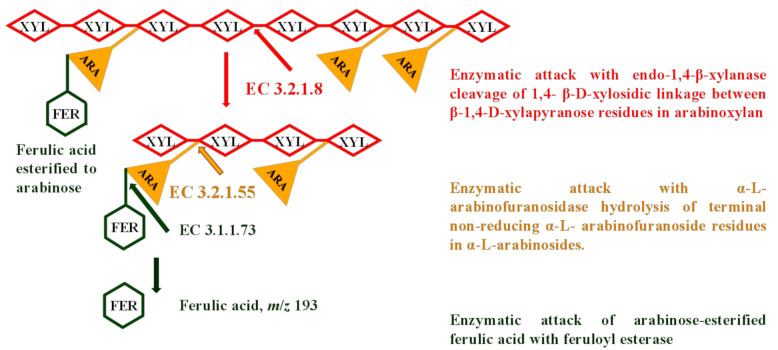
Simplified sketch of the enzymes’ synergistic effect in the hydrolysis of arabinoxylan found in cereal bran and involvement of them in the release of esterified ferulic acid (FER). The action of endo-1,4-β-xylanase results in the cleavage of internal 1,4-β-d-xylosidic linkages in β-1,4-d-xylapyranose residues of xylans (XYL). Hydrolysis of non-reducing end of terminal *α*-*L*- arabinofuranoside residues in *α*-*L*-arabinosides by *α*-*L*-arabinofuranosidase results in the release of feruloylated arabinose (ARA) monomers. The final attack of ester linkages between arabinose and ferulic acid with feruloyl esterase results in the release of free ferulic acid.

**Figure 2 foods-10-00782-f002:**
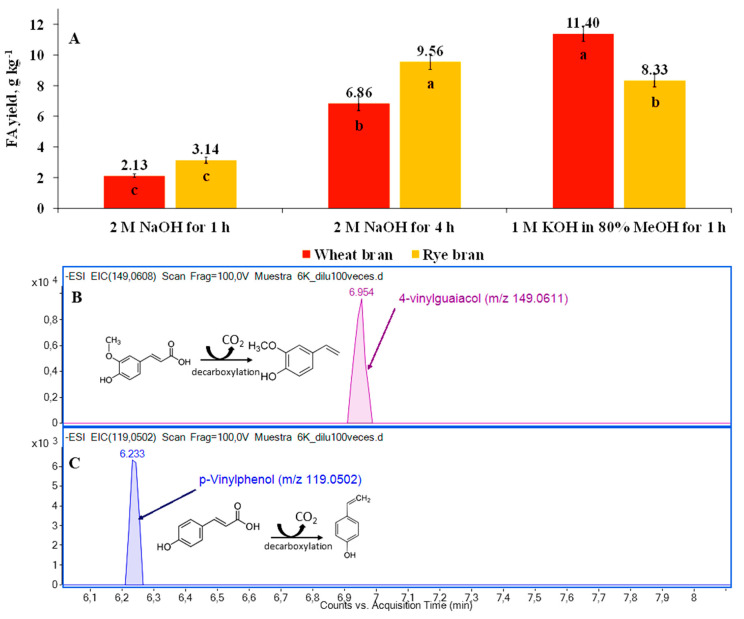
The recovery of ferulic acid (FA) using mild alkali-assisted hydrolysis of wheat and rye bran using either 2 M NaOH or 1 M KOH in 80% MeOH (**A**) and extracted ion chromatograms (EIC) plotted for the degradation products of FA–4-vinylguaiacol (**B**) and *p*-CA–*p*-vinylphenol (**C**). Note: Values are means ± SD values of triplicates. Means within the same bran type with different superscript letters (a, b, and c) are significantly different at *p* < 0.05. DW–dry weight.

**Figure 3 foods-10-00782-f003:**
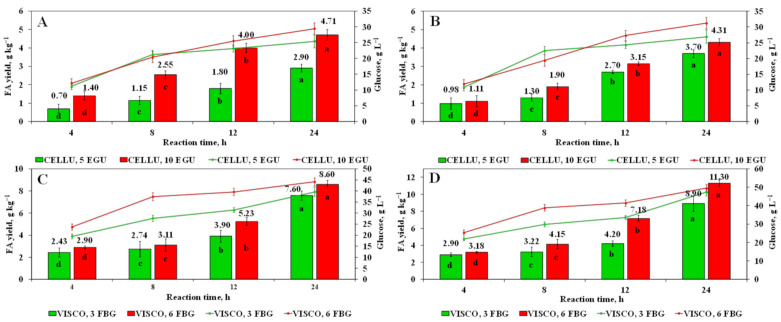
The recovery of FA using enzymatic hydrolysis of wheat (**A**,**C**) and rye (**B**,**D**) bran using either Celluclast^®^ 1.5 L and Viscozyme ^®^ L as a sole hydrolytic enzyme. Note: Values are means ± SD values of triplicates (*n* = 3). Means within the same type and dose of enzyme with different superscript letters (a, b, c, and d) are significantly different at *p* < 0.05. DW–dry weight. Red and green lines on the plots indicate the release of glucose in g L^−1^ of hydrolysates over 24 h enzymatic hydrolysis either with Celluclast^®^ 1.5 L and Viscozyme^®^ L as a sole hydrolytic enzyme.

**Figure 4 foods-10-00782-f004:**
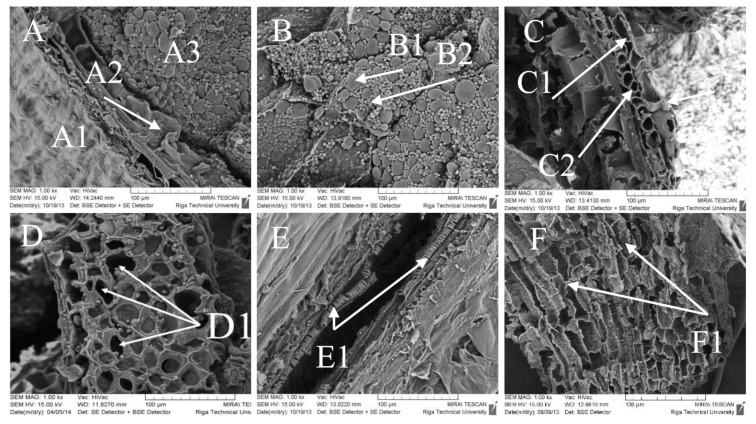
Micrographs from scanning electron microscope of the wheat grain (**A**) fractions: A1–Epidermis attached to aleurone layer A2 (proteins) and starchy endosperm A3 (starch); (**B)** starchy endosperm composed of large B1 (15–35 μm) and small B2 (smaller than 10 μm) starch granules; (**C**) aleurone layer filled with protein tingly attached to inner pericarp composed of tube cells (C1) and cross cells (C2); (**D**) Wheat bran after 24 h hydrolysis with multi-enzyme complex Viscozyme^®^ L with obvious signs of epidermal cracking (D1) (holes size of 20–30 μm); (**E**) Wheat bran after 4 h hydrolysis with multi-enzyme complex Viscozyme^®^ L with partially opened cellulose microfibers (E1); (**F**) Wheat bran after 24 h hydrolysis with multi-enzyme complex Viscozyme^®^ L with more tangible degradation of the epidermal layer (F1).

**Figure 5 foods-10-00782-f005:**
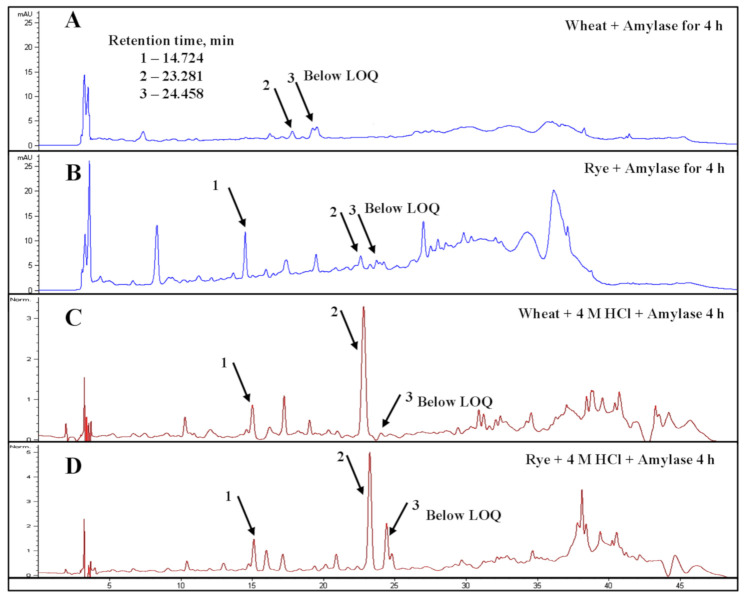
HPLC-DAD/MS^n^ chromatograms of the hydrolysates upon EH of wheat (**A**) and rye (**B**) bran with AMY for 4 h and digestates after acidic hydrolysis with 4 M HCl of wheat (**C**) and rye (**D**) bran for 4 h followed by EH accomplished by AMY within additional 4 h. Note: peaks 1, 2, and 3 correspond to caffeic, *p*-coumaric, and ferulic acids, respectively. Hydroxycinnamates were monitored at 320 nm in a negative ionization mode. The identification of compounds of interest was based on their retention times and mass spectra in full scan mode (MS).

**Figure 6 foods-10-00782-f006:**
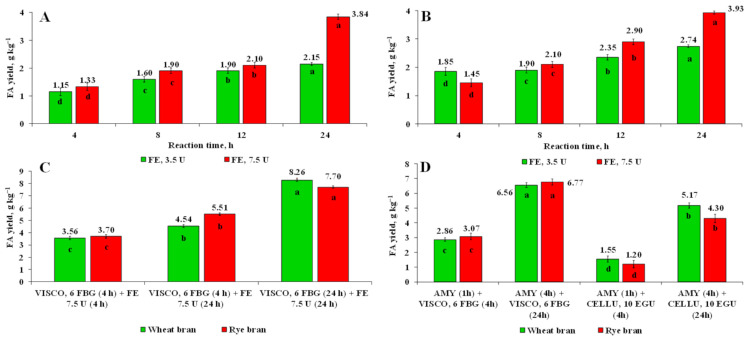
The recovery of FA using enzymatic hydrolysis of wheat (**A**,**C**) and rye (**B**,**D**) bran with feruloyl esterase as a sole enzyme or a mixture of multi-enzyme complex Viscozyme^®^ L coupled with feruloyl esterase or Amylase^®^ AG XXL in combination with either Viscozyme^®^ L or Celluclast^®^ 1.5 L. Note: Values are means ± SD values of triplicates (*n* = 3). Means within the same dose of feruloyl esterase (**A**,**B**) and the same type of bran (**C**,**D**) with different superscript letters (a, b, c, and d) are significantly different at *p* < 0.05. DW–dry weight.

**Table 1 foods-10-00782-t001:** Nutritional composition of bran by-products derived from wheat and rye grains, g 100 g^−1^ DW.

Major Nutrients Profile, g 100 g^−1^ DW.							
Type of Material	Moisture, %	Starch	Crude Lipids	Crude Proteins	Crude Cellulose	Crude HEM	ADL
Wheat bran	4.1 ± 0.1 ^a^	8.7 ± 0.3 ^b^	3.1 ± 0.0 ^b^	17.1 ± 0.8 ^a^	39.8 ± 4.1 ^a^	12.9 ± 0.9 ^a^	9.3 ± 0.0 ^a^
Rye bran	4.1 ± 0.1 ^a^	18.6 ± 0.1 ^a^	2.5 ± 0.0 ^a^	17.0 ± 0.7 ^a^	33.4 ± 2.1 ^b^	5.3 ± 0.5 ^b^	3.3 ± 0.0 ^b^

Note: Values are means ± SD values of three replicates (*n* = 3) of ten samples. Means within the same column with different superscript letters (a and b) are significantly different at *p* < 0.05. HEM–hemicellulose; ADL—acid detergent lignin; DW—dry weight.

**Table 2 foods-10-00782-t002:** The list of commercial hydrolytic enzymes used in this study.

Commercial Enzyme	Declared Activity	Enzyme Activity	Source	EC Number
Amylase^®^ AG XXL	460 AGU g^−1^	Glucan-1,4-α-glucosidase	*Aspergillus niger*	3.2.1.3
Celluclast^®^ 1.5 L	700 EGU g^−1^	1,4-β-D-endoglucanase	*Trichoderma reesei*	3.2.1.4
Viscozyme^®^ L	100 FBG g^−1^	Endo-1,4-β-xylanase α-L-arabinofuranosidase 1,4-β-D-endoglucanase	*Aspergillus aculeatus*	3.2.1.8 3.2.1.55 3.2.1.4
Megazyme™ Feruloyl esterase	30 U mg^−1^	Feruloyl esterase	Rumen microorganism, n.s.	3.1.1.73

Note: AGU—amyloglucosidase units; EGU—endoglucanase units; FBG—fungal β-glucanase units; n.s.—not specified.

## Data Availability

The data sets and analysis of current study are available from the corresponding author upon reasonable request.

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
