# Peer review of "Highly-Efficient Release of Ferulic Acid from Agro-Industrial By-Products via Enzymatic Hydrolysis with Cellulose-Degrading Enzymes: Part I–The Superiority of Hydrolytic Enzymes Versus Conventional Hydrolysis"

_foods, 2021, doi:10.3390/foods10040782_

Round 1
Reviewer 1 Report
Valorisation of cellulose containing biomass is one of the key issues of research and development activities. Development and investigation of the efficiency of method that is suitable to produce bioactive components and/or raw materials from by-product for food and pharmaceutical industry can provide interesting information for the science and practice, as well.
Therefore, the main concept of research presented in manuscript foods-1158166 is considered as interesting topic for the readers. Ferrulic acid extracted from wheat and rye bran is a valuable component for the food and pharmaceutical industry.
Research motivations are well defined, the manuscript is well structured.
In the study the experiments were conducted to compare the ferrulic acid recovery by using conventional hydrolysis and saponification and enzymatic hydrolysis. Materials and methods are given in details. Applied methods are adequate and accepted in the science and practice.
Manuscript contains interesting and valuable results that are discussed with relevant references. Results are discussed in details.
Comments, suggestions
As I know PVDF acronym is general used for polyvinylidene fluoride materials (line 177, 189).
How was the time of enzymatic hydrolysis determined? As Figure 3 show the FA yield increased during the 24 hrs time period. Is it possible to achieve higher yield by using longer enzymatic hydrolysis process?
Have the authors experienced inhibition effects during enzymatic hydrolysis?
Have the authors information related to the cost of different hydrolysis methods?
Author Response
R: Valorisation of cellulose containing biomass is one of the key issues of research and development activities. Development and investigation of the efficiency of method that is suitable to produce bioactive components and/or raw materials from by-product for food and pharmaceutical industry can provide interesting information for the science and practice, as well.
A: Dear reviewer!
We appreciate your time and contribution very much. Thank you also for your positive feedback on our manuscript.
R: As I know PVDF acronym is general used for polyvinylidene fluoride materials (line 177, 189).
A: Dear reviewer!
We fully agree with you that the PVDF acronym refers to polyvinylidene fluoride materials. It has been typing error in our manuscript which is now corrected.
R: How was the time of enzymatic hydrolysis determined? As Figure 3 show the FA yield increased during the 24 hrs time period. Is it possible to achieve higher yield by using longer enzymatic hydrolysis process?
A: We appreciate your valuable question regarding the yield of FA as a function time. Based on our observations, it should be noted that no significant effect of longer enzymatic hydrolysis has been observed towards the yield of FA. It has been decided to terminate the reaction after 24h of enzymatic hydrolysis since additional energy input can affect the price of the final product.
R: Have the authors experienced inhibition effects during enzymatic hydrolysis?
A: Dear reviewer! Many thanks for your practical question regarding the activity of enzymes used. It is worth noting that no delay in enzymes activity has been observed within 24 h of enzymatic hydrolysis. Our conclusion is based on the continuous increase of glucose in slurries as a function of hydrolysis time. However, after 24 h no tangible increase of FA has been observed. This observation makes it possible to conclude that the glucose and possibly other sugars accumulated during the final stage of EH could negatively affect the activity of hydrolytic enzymes. To validate this observation, a more detailed analysis of sugars, including arabinose, xylose, glycerol need to be done. These data will be shown in our next work.
R: Have the authors information related to the cost of different hydrolysis methods?
A: Dear reviewer! We appreciated your meaningful question. Unfortunately, we don't have such information. Currently, the experiments performed on hydrolysis of lignocellulosic materials do not go beyond the laboratories or are disseminated rather ineffectively. No calculated profitability or predicted prices for the final products obtained using this technology could be seen. The feasibility of this technology towards the reduction of environmental pollution has been widely documented, while no data available so far regarding specific costs for the production of a particular product.
Reviewer 2 Report
The manuscript explores the substitution of enzymatic hydrolysis (EH)together with biorefining to produce Ferulic Acid (FA) as more sustainable way to obtain this polyphenol in substitution of alkali hydrolysis, for wheat and rye by using 3 different enzymes and their combination to prove their efficacy in comparison to classical methods. Results has shown that EH, especially with Viscozyme resulted with equivalent or greater yield than the mild alkali hydrolysis with the advantage that FA is less degraded to 4-VG or p-CA by EH.
The manuscript doesn´t offer a great novelty with the substitution of enzymes, but proves the production of FA in a more sustainable way.
The conclusions are supported by the data and the article is clearly and logically written. The theme is consistent with the journal scope.
Some consideration should be taken in account and error have been found:
Table 1- n.d. doesn’t appear in the Table
Line 203- substitute “fermentation” by “enzymatic hydrolysis”
Line 368, please clarify if it is 1 m KPOH, 1 h or 2M KOH 1h (fig. 1)
Author Response
R: The manuscript explores the substitution of enzymatic hydrolysis (EH)together with biorefining to produce Ferulic Acid (FA) as more sustainable way to obtain this polyphenol in substitution of alkali hydrolysis, for wheat and rye by using 3 different enzymes and their combination to prove their efficacy in comparison to classical methods. Results has shown that EH, especially with Viscozyme resulted with equivalent or greater yield than the mild alkali hydrolysis with the advantage that FA is less degraded to 4-VG or p-CA by EH.
The manuscript doesn´t offer a great novelty with the substitution of enzymes, but proves the production of FA in a more sustainable way.
The conclusions are supported by the data and the article is clearly and logically written. The theme is consistent with the journal scope.
A: Dear reviewer!
We appreciate your time and contribution very much. Thank you also for your positive feedback on our manuscript.
R: Table 1- n.d. doesn’t appear in the Table
A: Dear reviewer! We appreciate your observation. No "n.d." appears anymore under Table 1.
R: Line 203- substitute “fermentation” by “enzymatic hydrolysis”
A: Dear reviewer. Many thanks for your valuable suggestion. The term "fermentation" has been substituted by "enzymatic hydrolysis"
R: Line 368, please clarify if it is 1 m KPOH, 1 h or 2M KOH 1h (fig. 1)
A: Dear reviewer. Many thanks for your note. In this experiment 1 M KOH in 80% MeOH has been used. Clarification has been provided in Fig. 2.